# Effect of High Sulfur Diet on Rumen Fermentation, Microflora, and Epithelial Barrier Function in Steers

**DOI:** 10.3390/ani11092545

**Published:** 2021-08-30

**Authors:** Hao Wu, Yan Li, Qingxiang Meng, Zhenming Zhou

**Affiliations:** 1State Key Laboratory of Animal Nutrition, College of Animal Science and Technology, China Agricultural University, Beijing 100193, China; wu2213@cau.edu.cn (H.W.); liyan@moahr.cn (Y.L.); qxmeng@cau.edu.cn (Q.M.); 2Human Resources Development Center of Ministry of Agriculture and Rural Affairs and China Association of Agricultural Science Societies, Beijing 100125, China

**Keywords:** beef cattle, high sulfur diet, rumen fermentation, microbiology, barrier function

## Abstract

**Simple Summary:**

Effect of high sulfur diet on digestion and morphology of the ruminant gastrointestinal tract was investigated both in vitro and in vivo. The results showed that, though sulfur level had little effect on rumen fermentation and most of the rumen microbials, sulfate-reducing bacteria (SRB) pop-ulation and sulfur metabolism had been changed, which led to inhibit methane emission. How-ever, high sulfur in the diet could increase risk of inflammation of rumen epithelium.

**Abstract:**

These experiments were conducted to evaluate the effect of excessive sulfur on rumen fermentation, microflora, and epithelial barrier function in steers through in vitro gas production and animal feeding experiments. Nine and four levels of sulfur addition were evaluated in in vitro ruminal fermentation and animal feeding experiment, respectively. The results showed that increasing the level of sulfur in substrates decreased the total gas and methane production linearly, while increasing the production of hydrogen sulfide gas (*p* < 0.01). Volatile fatty acid concentrations, especially that of butyric acid, were increased by extra sulfur (*p* < 0.01). Sulfur content in the diet had no significant effect (*p* > 0.05) on most of the rumen microbes, except for *Desulfovibrio*, one of the major sulfate-reducing bacteria (SRB) in the rumen, whose population increased by adding extra sulfur (*p* < 0.001). The changes in the morphology of rumen epithelium and thickening of the total epithelial layer were mainly attributed to the increase in the acanthosis cell layer and stratum basale (*p* < 0.05). Further, the relative expressions of two tight junction protein regulating genes, CLDN-1 and TJP1, were reduced (*p* < 0.05). Excessive sulfur in the diet can change the type of rumen fermentation, sulfate metabolism and SRB population, and the rumen epithelial barrier function. The results of this study demonstrated that sulfur can be used as a methane inhibitor with the mechanism that SRB competitively used protons to produce hydrogen sulfide. However, a higher level of sulfur in the diet could increase the inflammatory reaction of the rumen epithelium which may affect nutrient absorption.

## 1. Introduction

Sulfur plays an essential role in the metabolic, structural, and regulatory functions of all living organisms, because of the S-containing compounds, such as amino acids, hormones, B vitamins, and co-enzymes. In recent years, dried distiller grains with solubles (DDGSs) have become abundantly available and are cost-competitive as a feed ingredient in the ruminants because of the expansion in the bioethanol industry all over the world. However, sulfuric acid is added to remove the impurities in bioethanol processing, which increases the sulfate content in DDGSs, making it more sulfurous than corn. The level of sulfur can range from 0.3% to more than 1.0% [1]. Corn-based DDGSs contains an average of 30% crude protein (CP), with a ruminal degradability of 45%, and 11–17% of ether extracts [2], all of which make DDGSs a cost-competitive, alternative feed for ruminants. The National Research Council (NRC) recommends 0.15% of sulfur content in the beef cattle diet, and when the sulfur content is higher than its tolerance level of 0.4%, it reduces animals′ dry matter intake (DMI) and feed conversion efficiency, eventually limiting the amount of DDGSs that can be added to the feed.

Most of the dietary sulfur ingested by ruminant animals is converted to sulfide by rumen microbes, primarily the rumen bacteria. The S-containing amino acids are fermented to sulfide, while sulfate is also reduced to sulfide by ruminal sulfate-reducing bacteria (SRB) [3]. Although SRB are comprised of a small population of bacteria in the rumen, it can have a significant impact on the cattle fed with a high sulfur diet, such as DDGSs.

Increased DDGSs content in the diet also increases hydrogen sulfide (H_2_S) production by rumen SRB, increasing the risk of sulfide-induced polioencephalomalacia (PEM) [1]. The rumen epithelium plays an important role in the nutrient absorption and defense in ruminants. The normal barrier function of rumen epithelium is very important to ensure their health. However, it is unclear if H_2_S damages the rumen epithelium and affects the health of the animal, leading to an inflammatory response.

We hypothesized that a high concentration of sulfur in the diet could change the rumen fermentation and microflora, and the metabolic products of micro-organisms can further affect the barrier function of the rumen epithelium, causing inflammation. Therefore, our study aimed at exploring the effect of high-sulfur diet in rumen epithelial injury and inflammation using in vitro rumen cultures, microbiology, and histomorphology analysis, and to provide the fundamental information to clarify the interaction between dietary nutrition, rumen micro-organisms, and animal inflammation.

## 2. Materials and Methods

The experiments were conducted at the Beef Cattle Research Station of China Agricultural University in Beijing. Animal management and research procedures were approved by the Animal Welfare and Ethical Committee of China Agricultural University (Permit No. DK3178). The experiments were performed as per the Regulations of the Administration of Affairs Concerning Experimental Animals (The State Science and Technology Commission of China, 1988).

### 2.1. In Vitro Incubation Procedure

In total, nine levels of sulfur addition, including 0.4% (basal sulfur content of the substrate dry matter), 0.5% (using sodium sulfate as exogenous sulfur source), 0.6%, 0.7%, 0.8%, 0.9%, 1.0%, 1.1%, and 1.2% were evaluated with a complete random design. Fresh rumen fluid was collected from three Angus steers (350 ± 43 kg) having permanent rumen fistula, before their morning feed. They were fed a total mixed ration twice a day at 0800 and 1600 h, the fluid was then squeezed and filtered through four layers of cheese cloth into a vacuum bottle. The samples were transported immediately to the laboratory of Beef Cattle Research Center of China Agricultural University. The in vitro incubation was carried out according to the procedures of Menke et al. [4]. The rumen fluid was mixed with artificial saliva, which was made according to Menke et al. [4], in a 1:2 (*v/v*) proportion, under a continuous flux of CO_2_ to prepare in vitro fermentation inoculum.

Of feed samples, 220 mg, same as the ration fed to fistula cattle, with the right sulfur levels were transferred to 100 mL glass syringes (Deli Electric Power Equipment, Shijiazhuang, Hebei, China), pre-incubated at 39 °C, with each treatment containing nine syringes. Of the inoculum, 30 mL was injected into each syringe with a Varispenser (Eppendorf AG, Hamburg, Germany) and then incubated on an automatic shaker (Jie Cheng Experimental Apparatus, Shanghai, China), in a water bath at 39 °C for 72 h. During the incubation, the volume of cumulative gas production (GP) was recorded manually at the time points of 0, 2, 4, 6, 8, 10, 12, 18, 24, 30, 36, 42, 48, 60, and 72 h.

At the end of 24 h incubation, triplicate gas samples were collected from each syringe after terminating the fermentation process in ice water, to determine the CH_4_ and H_2_S concentration. CH_4_ was determined by Gas Chromatography (TP-2060F, Beijing Beifen Tianpu Analytical Instrument Co., Ltd., Beijing, China), and H_2_S was determined using GASTEC fast gas detection tube (4HH, Gastec Corporation, Kanagawa, Japan,). The fermentation mixture was sampled at 24 h and 48 h and then centrifuged at 8000× *g* for 15 min at 4 °C. Thus, the obtained supernatant was used to determine the volatile fatty acids (VFA) and ammonia-nitrogen (NH_3_-N) using methods described by Wu et al. [2].

### 2.2. Animals, Diets, and Experimental Design

Eight 24-month-old Angus steers (350 ± 43 kg) having permanent rumen fistula were selected as the experimental animals in a repeated 4 × 4 Latin square design. Four sulfur addition levels (0.4%, 0.6%, 0.8%, and 1.0%), with 0.2% as the interval, were set up according to orthogonal polynomial contrast analysis requirement. Each level was fed to two animals in a period of 20 days, where 18 days were meant for adaptation and two days for sample collection. The basic diet contained 0.4% sulfur(DM basis), which was determined using the Magnesium nitrate method described in GB/T 17776-1999 (National Standards of People’s Republic of China), and was used as a control (CON), while sodium sulfate (Na_2_SO4) was added as an exogenous sulfur source to provide the right amount of additional sulfur, which were set as low additional sulfur (LAS; 0.6% sulfur, DM basis), moderate additional sulfur (MAS; 0.8% sulfur, DM basis), and high additional sulfur (HAS; 1.0% sulfur, DM basis) treatments (Table 1).

### 2.3. Sample Collection and Measurement

#### 2.3.1. Collection of Rumen Fluid

On the first day of each sampling period, rumen content was collected through fistula before the morning feed at the following four sites: rumen sac, dorsal cecum, abdominal sac, and abdominal cecum. It was then mixed and filtered using four layers of cheesecloth, and transferred to 20 mL centrifuge tubes to further concentrate VFA and measure the NH_3_-N content. Additionally, 2 mL of rumen contents were cryopreserved in triplicate and immediately stored in liquid nitrogen for further DNA extraction and microbial analysis.

#### 2.3.2. Collection of Rumen Epithelial Tissue Samples

After the first day of sampling, the experimental animals were allowed to fast, with no solid and liquid intake for the next 24 h, and the rumen epithelial tissue was collected from the abdominal sac portion of the rumen. The sac was pulled toward the fistula by hand after emptying the remaining fistula contents. The chyme residue on the surface of the rumen epithelium was washed with saline, and then the tissue was cut thoroughly with a surgical scissor soaked in DEPC water, while the animal was under topical anesthesia. The tissue samples were cut into 1 × 1 cm and 0.5 × 0.5 cm pieces and placed in 4% polyformaldehyde (Sigma, St. Louis, MO, USA) and 2.5% glutaraldehyde solution for histomorphological analysis.

### 2.4. DNA Extraction, High-Throughput Sequencing, and Data Processing

DNA was extracted from 0.5 g of each of the homogenized ruminal semi-fluid samples using the repeated bead-beating plus column purification method [5] and an oscillator (Precellys 24, Bertin Technologies, Montigny-le-Bretonneux, France). The rotation speed of the oscillator was 5500 rpm with two circulations (30 s per circulation). The DNA quality was assessed by agarose gel (1%) electrophoresis, and metagenomic DNA concentrations were determined using a NanoDrop 2000 Spectrophotometer (Thermo Fisher Scientific, Waltham, MA, USA). Further, DNA was diluted to a concentration of 1 ng/uL using sterile water.

Sequencing was performed on an Illumina MiSeq PE300 platform. DNA was amplified using the universal eubacterial primer set (338F: 5′-ACTCCTACGGGAGGCAGCAG-3′ and 806R: 5′-GGACTACHVGGGTWTCTAAT-3′), which targets the hypervariable V3-V4 region of the 16S rRNA gene. The reverse primer contained a 6-bp error-correcting barcode, unique to each sample [6] and the 5‘-end of the reverse primer was fused to an Ion A adaptor plus a key sequence and a sample barcode sequence. Whereas the forward primers were fused to a truncated Ion P1 adapter sequence. PCR conditions and the amplification reaction protocol have been previously described by Zhou et al. [7]. Amplicons were examined on a 2% E-Gel Size SelectTM Agarose Gel and purified with Agencourt AMPure XP Reagent. Library size and molar concentration were determined by Agilent 2100 BioanalyzerTM using Agilent High Sensitivity DNA Kit (Agilent Technologies, Inc., Santa Clara, CA, USA). Emulsion PCR was performed using the Ion OneTouchTM 200 Template Kit v2 DL (Life Technologies, Inc., Delhi, India) according to the manufacturer’s instructions. The sequencing of the amplicon libraries was performed on a 318 chip by the Ion Torrent Personal Genome Machine (PGM) system using the Ion PGMTM Sequencing 300 kit (Life Technologies, Inc., Delhi, India).

The Illumina MiSeq sequencing data were analyzed using QIIME software (version 1.7.0) [8]. Filters were applied to the sequences before phylogenetic analysis. Depending on the appropriate fragment size for V3-V4 PCR (150–200 bp), bases after the 200th position were trimmed and the reads shorter than 150 bp were removed. Reads with a quality score of <25 were removed using the NGS QC Toolkit, and only sequences without any ambiguous characters were included in the analysis. FLASH 1.2.7v was used to merge the paired-end reads from raw sequencing data [9]. Chimeric sequences were removed using USEARCH based on the UCHIME algorithm [10]. In order to calculate the downstream diversity (alpha and beta diversity), all the samples were subsampled in equal sizes of 100,000 reads before comparing the bacterial communities. The sequences were clustered into operational taxonomic units (OTUs) at 97% sequence identity level, and the most abundant sequence from each OTU was chosen as a representative. Based on the OTUs, rarefaction curve and alpha diversity indices (i.e., abundance-based coverage estimator (ACE), and the Chao 1, Shannon, and Simpson estimators) were developed. The jackknifed beta diversity was visualized by Principal component analysis (PCA) and performed using the UnscramblerX program (CAMO Software Inc., Woodbridge, NJ, USA) to identify the shifts in the microbial population structure.

### 2.5. Pretreatment of the Rumen Epithelial Tissue Section for Staining and Electron Microscopy

Ten papillae per animal were prepared for light microscopy histomorphometric analysis using methods previously described by Odongo et al. [11]. PFA-fixed, paraffin-embedded papillae were sectioned at 6 µm thickness, stained with hematoxylin and eosin, and mounted for analysis. The microscopist was blinded to the treatment conditions during the histomorphometric analysis. Measurement of each stratum was made using the 40× objective lens, and four images were captured per papillae for a total of 40 replicates per measurement per animal. Image-Pro Plus software (Media Cybernetics, Bethesda, MD, USA) was used to measure the predefined criteria previously described by Steel et al. [12].

Additional papillae were prepared for electron microscopy using methodology reported by Graham and Simmons [13]. Washed papillae were immediately fixed in 2.5% glutaraldehyde for 24 h, then fixed in 1% osmium for 1 h, and dehydrated in a series of graded ethanol solutions. For scanning electron microscopy (SEM), the papillae were subjected to critical-point drying using liquid CO_2_ as a medium, then mounted, and coated with gold. The samples were then examined using SEM (Hitachi Model S-3000N, Hitachi Technologies, Tokyo, Japan). For transmission electron microscopy (TEM), dehydrated samples were placed in a mixture of Spurr resin and acetone (1:1) for 30 min, followed by 10 h in 100% resin. These samples were placed in molds and polymerized at 40 or 60 °C for 48 h. Semithin (0.25–0.5 µm) sections were cut with glass knives and stained with 1% toluidine blue-O in 1% sodium borate. Ultrathin (70–90 nm) sections were cut using a diamond knife, stained with methanolic uranyl acetate followed by lead citrate, and examined using TEM (Hitachi H-7650, Hitachi Technologies, Tokyo, Japan) [14].

### 2.6. Measurement of Relative Expression of RNA

#### 2.6.1. RNA Isolation and cDNA Synthesis

Total RNA was extracted from the papillae samples using Trizol as described by Chomczynski and Sacchi [15]. The RNA concentration was quantified using a Nanodrop spectrophotometer ND-1000UV-Vis (Thermo Fisher Scientific, Madison, WI, USA). The absorption ratio (260/280 nm) of all the samples was between 1.96 and 2.09, indicating a high RNA purity. Aliquots of RNA samples were subjected to electrophoresis using 1.4% agarose-formaldehyde gel to verify its integrity. The concentration of RNA was adjusted to 1 µg/µL based on the optical density and then stored at −80 °C. Total RNA (1 µg) was reverse-transcribed using a PrimeScript^®^ RT reagent Kit with gDNA Eraser (Takara Bio Inc., Beijing, China) according to the manufacturer’s instructions.

#### 2.6.2. Primer Design and qRT-PCR

Primer sets were designed to recognize and amplify the conserved nucleotide sequences encoding bovine TJ proteins and cytokines. cDNA sequences were identified using BLAST (Basic Local Alignment Search Tool) (National Center for Biotechnology Information, Bethesda, MD, USA) and primers were designed using the Primer 5.0 (Whitehead Institute, Cambridge, MA, USA). All the primers were synthesized by Genenode Biotechnologies (Beijing, China). Real-time quantitative PCR of target genes and β-actin was performed using the ABI 7500 real-time PCR system (Applied Biosystems, Foster, CA, USA) by detecting the fluorescence of SYBR green dye. Amplification conditions were as follows: A temperature of 94 °C for 2 min, followed by 94 °C for 15 s, 60 °C for 15 s, and 72 °C for 30 s for 30 cycles. Each sample contained 2 µL of cDNA in 5 µL 2×SYBRGreen PCR Master Mix with 10 µM of each primer in a final volume of 10 µL. All the measurements were performed in triplicate. A reverse-transcription-negative blank for each sample and a no-template blank were used as negative controls. The relative amount of each studied mRNA was normalized to the mRNA level of the housekeeping gene, β-actin, and the data were analyzed according to the 2^–^^△△^^CT^ method. The primers and amplicon sizes of all the genes are presented in Table 2.

### 2.7. Statistical Analysis

In order to estimate the kinetic parameters of Cumulative Gas production (GP), all the results of GP were fitted using the NLIN Procedure of the SAS 9.0 statistical software. The formula used is as follows: a = b × (1 − e^−ct^), where: a = volume (mL) of gas production per 0.2 g DM substrate at time t; b = asymptotic gas production (mL) of 0.2 g DM substrate; c = rate of gas production per hour. The test data were preliminarily sorted by Excel, and the one-way ANOVA was performed by SAS 9.0 statistical software. Duncan’s multiple comparison test was used to calculate the SEM and *p* Values. Additionally, the linear and quadratic curve trend analysis (Orthogonal polynomial contrast) was carried out. *p* < 0.05 (significant) and *p* < 0.01 (extremely significant) were used as the criteria to judge the significance of differences.

## 3. Results

### 3.1. Effect of High Sulfur Diet on Rumen Fermentation

As shown in Figure 1, all nine treatment groups showed the same GP pattern. There was no lag at the beginning of the fermentation, and for the first 10 h, GP was the same among all the treatments, but with the extension of fermentation time in vitro, the GP rates changed, and the curve reached a stable phase after 48 h of incubation. According to the data shown in Table 3, the GP at 24 and 48 h in all nine treatment groups was significantly different (*p* < 0.01), which later decreased linearly with increasing sulfur content in the substrates. An increasing dose of sulfur reduced methane production but increased the production of hydrogen sulfide (Figure 2). Methane in the headspace was decreased from 19% to 16% with an increase in the sulfur content from 0.0% to 1.2% (DM basis), respectively. Meanwhile, at different doses of sulfur, the accumulation of hydrogen sulfide gas in the culture headspace increased linearly from 3500 to 6100 ppm.

Table 3 and Table 4 show the effects of sulfur on rumen GP and fermentation parameters (NH_3_-N and VFA). With an increase in the sulfur content of the substrate, the theoretical maximum gas production (B value) in the in vitro fermentation decreased linearly (*p* < 0.01), while the gas production rate (c value) increased linearly (*p* = 0.01). There was no effect of increased doses on NH_3_-N and VFA concentrations at 24 and 48 h except for the total volatile fatty acids (TVFA), which showed an increase (*p* < 0.05) at 24 h in vitro. Intriguingly, in vivo fermentation parameters were significantly affected by the addition of sulfur in the diet (*p* < 0.05). The concentration of NH_3_-N decreased significantly in LAS and MAS groups compared to the CON and HAS groups (*p* < 0.01). The concentration of TVFA in HAS was significantly higher than the other groups (*p* < 0.01). Moreover, the individual VFA concentration was significantly different in increased sulfur concentrations. For example, feeding the ruminants with a higher level of sulfur increased the content of butyric acid (*p* < 0.01), but significantly decreased the content of other VFAs (*p* < 0.01).

### 3.2. Effect of High Sulfur Diet on Rumen Microflora

Sequencing produced 1,780,477 reads of the 16S rRNA genes from 32 samples, with an average of 55,640 sequences per sample, and a total of 2332 OTUs. In order to ensure adequate sampling while investigating the global community and species level changes across the diets, Good’s coverage test was performed, where the lowest estimate value was 0.982. It suggested that the sequencing depth enables the characterization of 98.2% of the rumen bacterial population. The other richness estimates and diversity indices were calculated as well (Table 5), and the results showed that the difference in the sulfur content of diet had no significant effect on the alpha diversity index of rumen microbial community (*p* > 0.05). Figure 3 presents a comparison study between 32 samples by principal component analysis (PCA), which shows that the LAS group was not separable from the other groups, but the CON group was distinguishable from the MAS and HAS groups.

According to the phylum level of comparison between the four bacterial community groups in the rumen (Figure 4), Bacteroidetes and Firmicutes comprise of the major phyla, accounting for about 50% and 35% of the total bacterial population, respectively, while Proteobacteria and other phyla account for a small proportion. At the genus level, Prevotella-1, Rikenaceae-RC9-gut-group, and Bacteroidales-S24–7-group constitute the major population of about 20%, 10%, and 10%, respectively. There were some significant differences in the relative abundance of the top 10 bacteria at different levels (Figure 5). At the phylum level, the relative abundances of Proteobacteria, Tenericutes, and Saccharibacteria were significantly different in each group (*p* < 0.05), and at the genus level, the relative abundance of Lachnospiraceae-NK3A20-group was significantly different in each group (*p* < 0.001).

The relative abundance of *Desulfovibrio* in the rumen is lower than 1%, and the difference between the groups is shown in Figure 6. The relative abundance of *Desulfovibrio* in rumen had significantly increased with the increase in sulfate content (*p* < 0.001).

### 3.3. Effect of High Sulfur Diet on Rumen Epithelial Barrier Function

Figure 7 shows different cell layers of rumen epithelium observed under a 4 × 10-fold light microscope. Image-Pro Plus 6.0 image analysis software was used to measure the thickness of different cell layers in the rumen epithelium. The difference between the rumen epithelium length in the four additional sulfur levels was noted (Table 6). The length of papillae in the rumen epithelium increased significantly with the increase in sulfur content (*p* < 0.01). The total epithelial thickness of the rumen epithelium in the HAS group was significantly higher than that of the other groups, and it gradually increased with the increase in sulfur concentration (*p* = 0.01). Although the thickness of the cuticle and granular layer was not affected (*p* = 0.17), the thickness of the spinous layer and basal layer increased significantly (*p* = 0.02).

As shown in Figure 8, each group was photographed at 50, 150, and 2500 times the field of vision, consecutively. Rumen papilla of the CON group showed a complete cuticle when observed under electron microscopy. Compared to the CON group, the degree of desquamation, indentation, and keratinization on the surface of rumen papilla in the HAS group was more obvious. Simultaneously, serious damage and incomplete keratinization occurred on the surface of the papillae, showing extensive exfoliation of the cuticle (Figure 8A,D,G,J). Additionally, a high sulfur diet can increase the contact area between rumen epithelium and micro-organisms, making more micro-organisms appear on the surface of the rumen epithelium (Figure 8C,F,I,L). Thus, microbial colonization is more obvious. The results showed significantly different microbial morphology on the surface of rumen epithelium in different treatment groups. The micro-organisms in the control group were mostly spherical and rod-shaped, while in the high sulfur diet group, they were sickle-shaped and appeared as actinomycete (Figure 8I,L).

Figure 9 shows different types of lesions on the surface of the rumen papilla in the HAS group, as observed in the SEM images. The epithelial keratinocytes of rumen epithelium in the HAS group were exfoliating, necrotic (Figure 9A,B, shown by arrows), budding (Figure 9C, shown by arrows) and forming large fissures (Figure 9D, shown by arrows).

For TEM, each group was photographed at 5000, 10,000, and 40,000 times the field of vision, consecutively. It was clearly shown that the number of tight junctions between the epithelial cells in the CON group was large, and also the structure of tight junctions was complete with obvious welding lines and clear cell boundaries (Figure 10A–C). With the increase in sulfur concentration, the number of tight junctions between the cells decreased, the arrangement was disordered (yellow arrows), the gap between the cells became larger, the boundary became blurred, and a large number of vacuoles appeared at the junction between the cells (purple arrows).

The relative expression of tight junction proteins TJP1 and CLDN-1 genes in rumen epithelial cells was significantly lower in the HAS group than in the CON group (*p* < 0.05). However, there was no significant difference in the relative expression of OCLN, CLDN-4, and CLDN-17 genes between the two groups (Figure 11).

## 4. Discussion

### 4.1. Effect of High Sulfur Diet on Rumen Fermentation

Carbohydrates in the feed can be fermented by rumen microorganisms to produce a large amount of VFA, CO_2_, and H_2_. Methanogenic bacteria use H_2_ and CO_2_ as important energy sources producing CH_4_, which accounts for 82% of the total amount of CH_4_ produced in the rumen. Excessive CH_4_ production pollutes the environment and also results in energy loss of the feed. These experimental results indicate that sulfur can inhibit CH_4_ production in rumen fermentation in vitro. Thus, it can be used as an alternate CH_4_ inhibitor for further studies in the future. However, the effects of SRB and H_2_S production on animal health should also be considered, and an appropriate concentration of sulfur, which reduces CH_4_ emission, should be explored. When SRB and methanogenic bacteria coexist, they compete for the common substrate H_2_ [16], indicating a competitive relationship between SRB and methanogenic bacteria. Our results show that sulfur level of the fermentation substrate affects the gas composition of fermentation in vitro and are in general agreement with the studies of Hunerberg et al. [17] and Wu et al. [2], who reported a 19.5% and 59.3% decrease, respectively, in CH_4_ emissions by feeding the beef cattle with different sulfur level diets.

A suitable nitrogen:sulfur (N:S) ratio in the diet is necessary to synthesize microbial proteins in the rumen and improve digestion and metabolism of cellulose. Adding sulfur to the diet can improve the efficiency of nitrogen utilization, but excessive addition can reduce nitrogen depositions [18]. According to the results of fermentation in vivo, the content of NH_3_-N decreased significantly after increasing the sulfur concentration, but then it started increasing gradually, which is consistent with the previous research. This experiment shows that an increase in sulfur content initially increases the utilization rate of NH_3_-N, but a continuous increase in sulfur concentration reduces the utilization rate of NH_3_-N, imparting a negative effect.

The main source of VFA in the rumen is microbial fermentation of cellulose and hemicellulose, which is accompanied by hydrogen production [19]. In the process of sulfate reduction, hydrogen is consumed [20], alleviating the inhibitory effect of hydrogen on fiber degradation and promoting the production of VFA. The animal fermentation test results show that the content of TVFA in a high sulfur diet is significantly increased, which is consistent with the above theory. The ratio of acetic acid to propionic acid in the rumen also fluctuated and changed significantly, suggesting that the high sulfur diet affected the fermentation mode of the rumen. Butyric acid stimulates the maturation of rumen epithelium and adapts to the environmental changes, but excessive butyric acid can cause excessive keratinization of rumen epithelium and damage its barrier function [21]. The results of this study show that a high sulfur diet significantly increased the concentration of butyric acid, which also corresponds to its effect on rumen epithelial barrier function.

### 4.2. Effect of High Sulfur Diet on Rumen Microflora

At the phylum level, Bacteroidetes and Firmicutes are the major bacteria in the rumen. Firmicutes degrades fibers and cellulose [22], while Bacteroidetes digests carbohydrates and ferments organic substances [23] and account for the highest proportion of rumen bacteria, which is consistent with the previous studies [24,25]. Studies have shown that metabolic disorders are associated with an increase in Proteobacteria. Thus, it can be used as a potential criterion to identify microecological disorders and diseases [26]. It could be inferred that with the increase in sulfur concentration, the relative abundance of Proteobacteria in the rumen increases, increasing the possibility of metabolic disorders.

Few studies have investigated the function of NK3A20 in the rumen, but they have used it clinically as a marker of immunity [27], colon cancer [28], and fatty liver [29]. Our results indicate that a high sulfur diet increases the content of NK3A20, which is speculated to increase the potential incidence of some diseases.

SRBs are a group of bacteria with similar functions, which obtain energy by dissimilating the sulfate reduction pathway while producing a large amount of sulfide. In this experiment, *Desulfovibrio* was detected at the level of genes, and was significantly different between the different groups (*p* < 0.05), indicating that a high sulfur diet leads to an increase in *Desulfovibrio* population in the rumen fluid. Although the relative abundance of SRB in the rumen is low (<0.4%), the accumulation of toxins in the rumen epithelium may cause inflammation, which is consistent with the result of morphological observation. Considering the relationship between SRB and various intestinal diseases, it can be inferred that increasing sulfur content in the diet may cause potential health hazards, leading to gastrointestinal or metabolic diseases.

### 4.3. Effect of High Sulfur Diet on Rumen Epithelial Barrier Function

The morphological structure and development of rumen epithelium are affected by individual as well as dietary factors. Some studies have shown that animals respond adaptively to the rumen produced VFA by increasing its absorption area to further absorb VFA, thereby regulating its VFA content [30]. Our results showed that a high sulfur diet could increase the length of rumen papilla due to the increased rumen VFA, eventually leading to the adaptive growth of rumen papilla. This also explains the variation of rumen VFA content.

A healthy, stratified, and flat epithelium of the rumen is usually covered by keratinocytes, which acts as a primary physical barrier to prevent the lower rumen epithelial cells from being invaded by toxic substances. Our results show that a high sulfur diet can cause excessive keratinization of rumen epithelium and cuticle accumulation, which changes the function of the epithelial barrier and reduces the short-chain fatty acid transport [31]. This may be related to the high concentration of propionic acid and butyric acid in the rumen that promotes excessive proliferation or differentiation of the rumen epithelial cells [32]. According to our results, the concentration of propionic acid and butyric acid in the rumen increased significantly with high sulfur diet, which is one of the reasons for rumen epithelial keratosis.

We observed excessive keratinization, mass exfoliation, necrosis, budding, and fissure of the rumen epithelium (scanning electron microscopy results). These changes may increase the entry of pathogenic bacteria and the toxic substances inside the body through the rumen wall, thereby damaging the physiological functions of the body [33]. Therefore, we further studied the effect of high sulfur diet on the ultrastructure of rumen epithelium using transmission electron microscopy. The results showed that a high sulfur diet could degrade the junction between epithelial cells and blur the boundary. Earlier studies on rumen epithelial damage have shown that under a short-term dietary change and stimulation, the body adapts and shows an incomplete keratosis of rumen epithelium with no cell scattering and necrosis [34]. The reason for the serious reaction in our results may be because of the long feeding time and the high sulfur content in treatment groups. A high sulfur diet results in the damage of the morphology and structure of the rumen epithelium, which is mainly manifested by excessive keratinization of the rumen epithelium, exfoliation of keratinocytes, degradation of intercellular junctions, and cell necrosis. These changes in morphology and structure may cause bacteria and toxic substances to enter the lamina propria of the rumen epithelium or even the blood, inducing the occurrence of diseases such as rumen inflammation.

In ruminants, the barrier function of rumen epithelium has a great influence on its normal functioning as well. However, there are only a few reports on the molecular mechanism of its barrier function. There are four types of cell junctions: tight junction, gap junction, adhesion junction, and desmosome junction. Among these, tight junction proteins, including Claudin-1, Claudin-4, Claudin-17, and Occludin, play an important role in maintaining the integrity of the rumen epithelium. The studies, focusing on sub-acute ruminal acidosis (SARA), have mostly investigated the effects rendered on the permeability and tight junctions of the rumen epithelium. Some studies have also shown that [35] induction of SARA can destroy the integrity of the rumen epithelium morphology and structure, thereby increasing its permeability, and eventually damaging the normal barrier function [36]. The underlying mechanism is related to the down-regulation of the expression of ZO-1, Claudin-1, and Occludin in rumen epithelium. These results are consistent with our results. Therefore, it is speculated that a high sulfur diet acts as stimulation to the rumen epithelium producing a response similar to SARA, where the expression of tight junction genes in the cells is down-regulated, destroying the function of the rumen barrier.

## 5. Conclusions

High sulfur diet can affect the normal fermentation of rumen, with little effect on the whole rumen microbiota and a specific effect on the population of *Desulfovibrio*. A high sulfur diet also destroys the integrity and barrier function of rumen epithelium, increases its permeability, and reduces the relative expression of tight junction genes.

## Figures and Tables

**Figure 1 animals-11-02545-f001:**
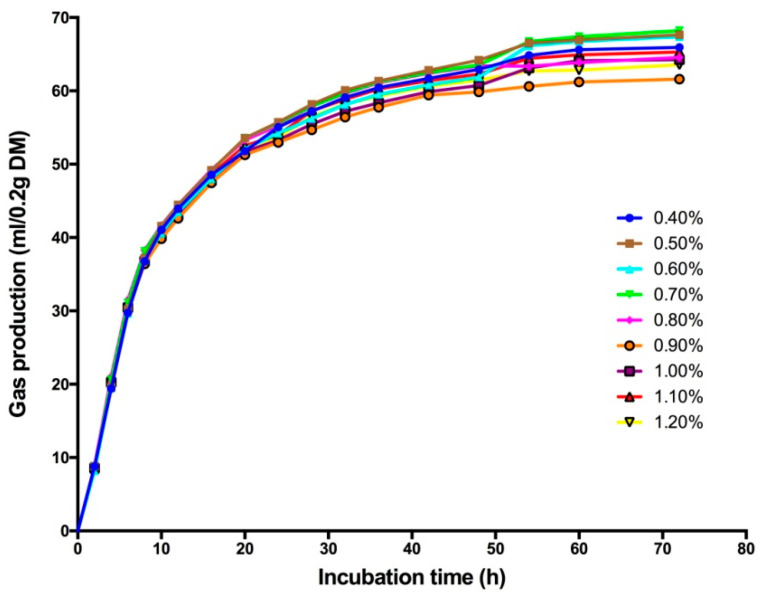
Dynamic changes in the gas production of in vitro ruminal fermentation.

**Figure 2 animals-11-02545-f002:**
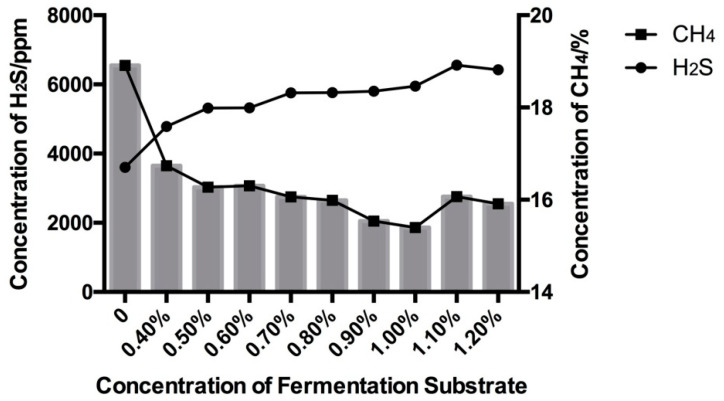
Gas content of in vitro fermentation of substrate with different levels of sulfate.

**Figure 3 animals-11-02545-f003:**
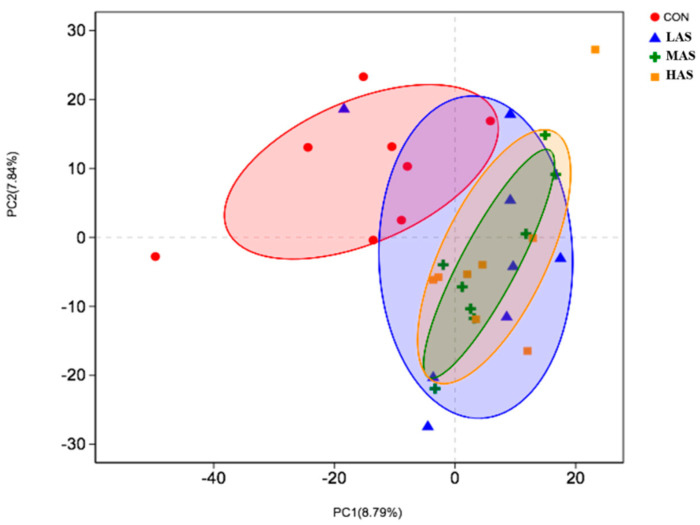
Principal component analysis of bacterial communities in different groups. CON-S (%DM) 0.4%; LAS-S (%DM) 0.6%; MAS-S (%DM) 0.8%; HAS-S (%DM) 1.0%.

**Figure 4 animals-11-02545-f004:**
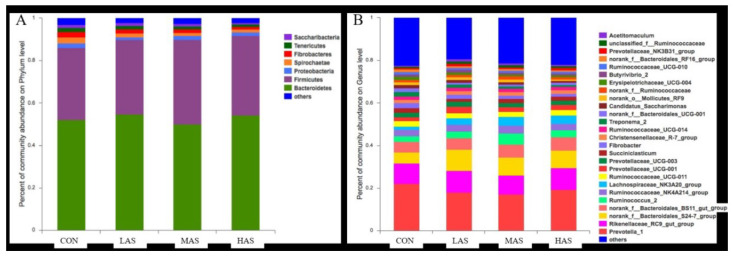
Comparison of the relative abundance of bacterial communities in the phylum (**A**) and genus (**B**) level.

**Figure 5 animals-11-02545-f005:**
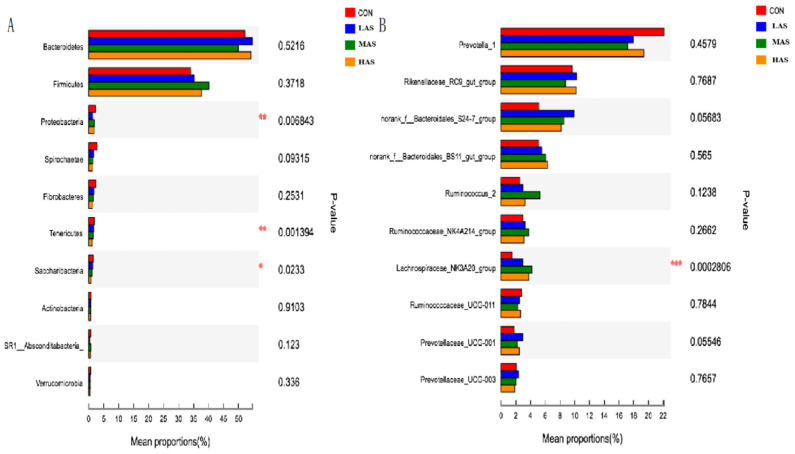
Difference test of bacterial communities in the phylum (**A**) and genus (**B**) level in different groups. CON-S (%DM) 0.4%; LAS-S (%DM) 0.6%; MAS-S (%DM) 0.8%; HAS-S (%DM) 1.0%. Asterisk means significant difference between treatments.

**Figure 6 animals-11-02545-f006:**
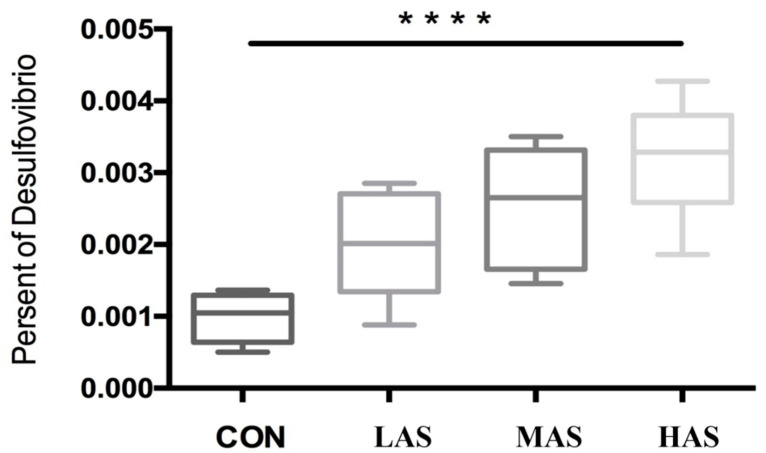
Present of *Desulfovibrio* in rumen samples. CON-S (%DM) 0.4%; LAS-S (%DM) 0.6%; MAS-S (%DM) 0.8%; HAS-S (%DM) 1.0%. Asterisks mean significant difference between CON and HAS groups.

**Figure 7 animals-11-02545-f007:**
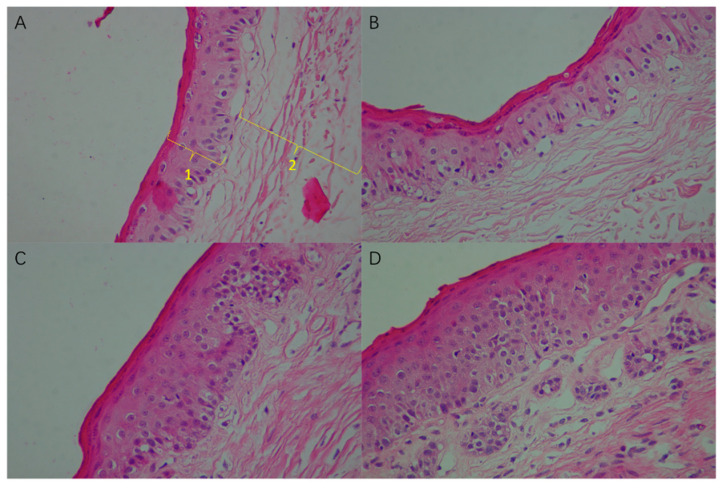
Effect of high sulfur diet on the morphology of rumen papilla (4 × 10 multiple microscope). (**A**), CON group; (**B**), LAS group; (**C**), MAS group; (**D**), HAS group. Epithelium cells (the black spots and red edge) arranged orderly (**A**,**B**) but became messy (**C**,**D**) with the increasing sulfur addition levels, which lead to increase the length of the epithelium. 1, cuticle and granular layer; 2, spinous layer and basal layer.

**Figure 8 animals-11-02545-f008:**
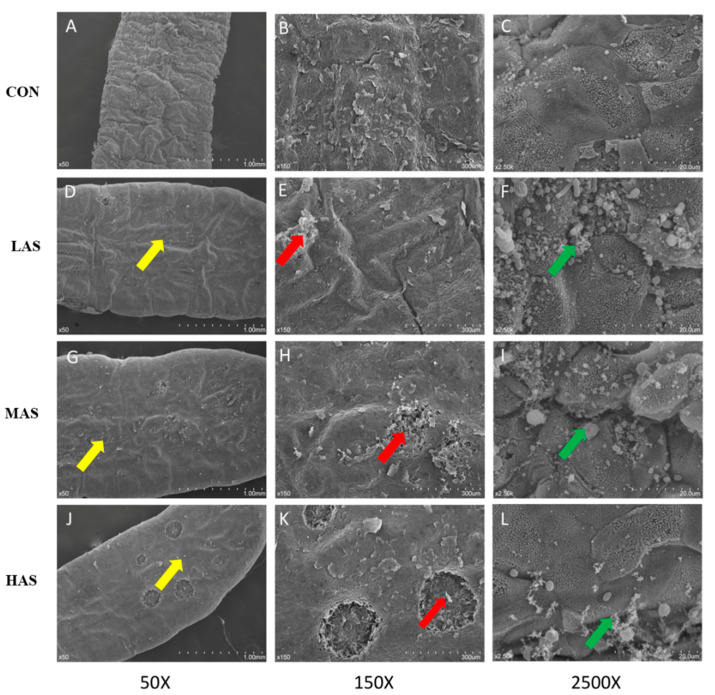
Scanning electron microscopy of papillae surface of different groups. CON-S (%DM) 0.4%; LAS-S (%DM) 0.6%; MAS-S (%DM) 0.8%; HAS-S (%DM) 1.0%. Cuticle exfoliation was more obviously (**A**,**D**,**G**,**J**, shown as yellow arrows). Serious damage occurred on the surface of papillae (**B**,**E**,**H**,**K**, shown as red arrows). More microbials were colonized (**C**,**F**,**I**,**L**, shown as green arrows).

**Figure 9 animals-11-02545-f009:**
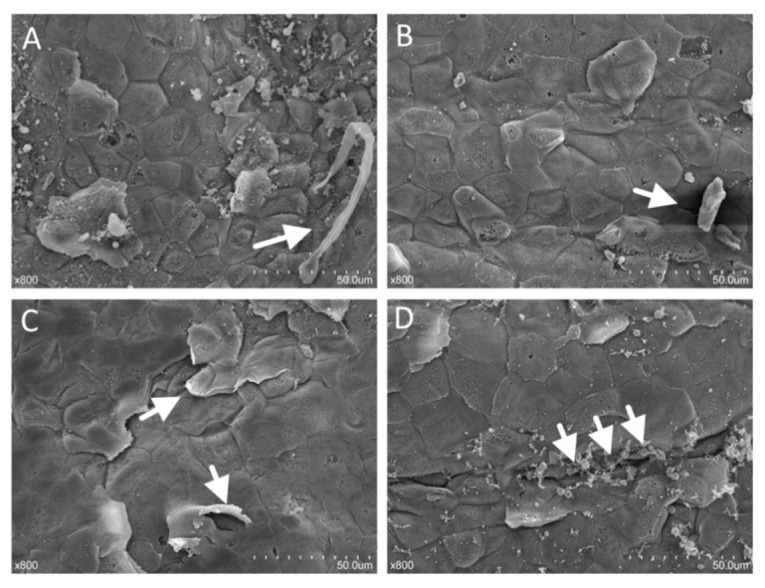
Different forms of stratum corneum cells disruption in the HAS group. (sloughing of the corneum, crack and budding). The epithelial keratinocytes of rumen epithelium were exfoliating, necrotic (**A**,**B**, shown by arrows), budding (**C**, shown by arrows) and forming large fissures (**D**, shown by arrows).

**Figure 10 animals-11-02545-f010:**
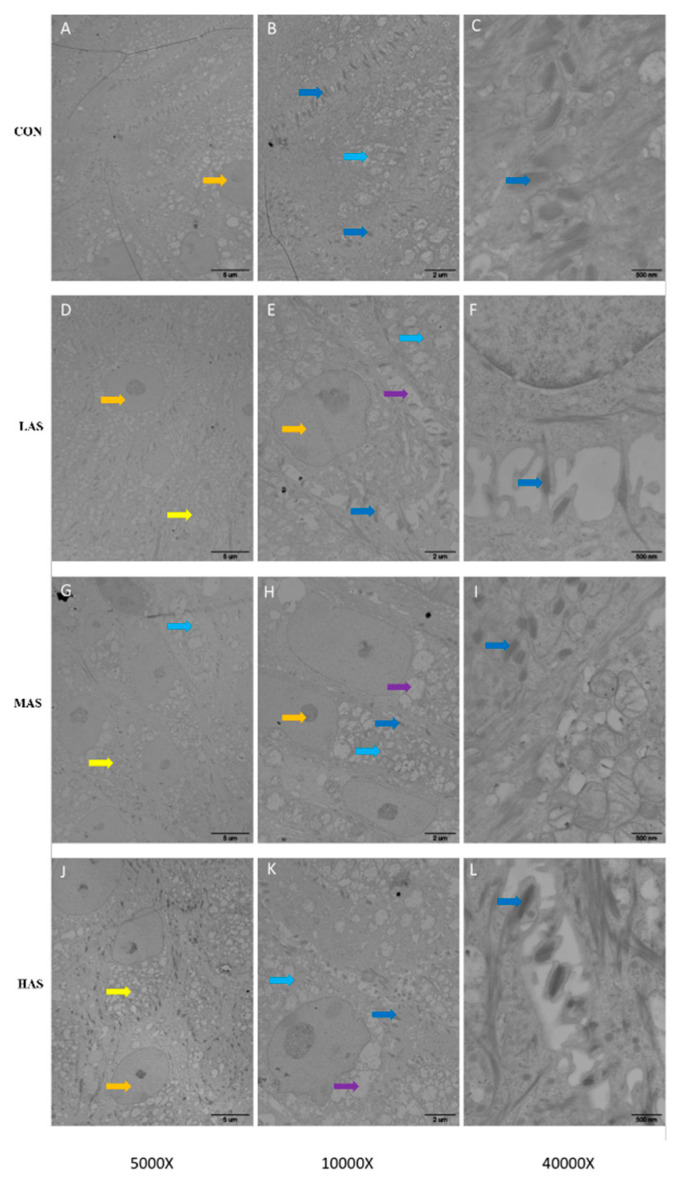
Effect of high sulfur diet on the ultrastructure of rumen epithelium under 5000, 10,000 and 40,000 times of TEM (**A**, **B**, **C** for CON; **D**, **E**, **F** for LAS; **G**, **H**, **I** for MAS; **J**, **K**, **L** for HAS, respectively). CON-S (%DM) 0.4%; LAS-S (%DM) 0.6%; MAS-S (%DM) 0.8%; HAS-S (%DM) 1.0%. Epithelium cells (orange arrows), tight junctions with welding lines (blue arrows) and other cell junctions (light blue arrows).

**Figure 11 animals-11-02545-f011:**
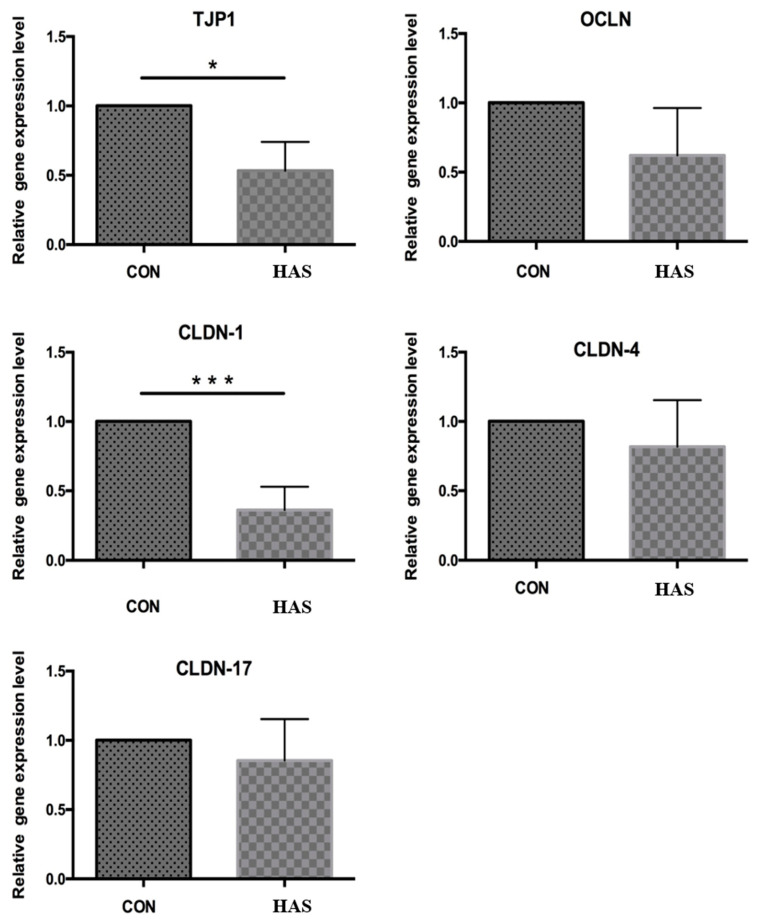
Effect of high sulfur diet on the relative expression of tight junction genes in the rumen epithelium. CON-S (%DM) 0.4%; HAS-S (%DM). TJP1 = tight junction protein 1; OCLN = Occludin; CLDN-1 = Claudin-1; CLDN-4 = Claudin-4; CLDN-17 = Claudin-17. Asterisks mean significant difference between treatments.

**Table 1 animals-11-02545-t001:** Basic diet composition and nutrition levels.

Item	Treatment ^1^			
	CON	LAS	MAS	HAS
Ingredient (%DM)				
Steam-flake corn	40.00	40.00	40.00	40.00
Whole corn silage	40.00	40.00	40.00	40.00
Cottonseed meal	4.50	4.50	4.50	4.50
Brewer’s grain	13.40	13.40	13.40	13.40
Sodium sulfate	-	0.89	1.78	2.66
Salt	0.50	0.50	0.50	0.50
Premix ^2^	0.50	0.50	0.50	0.50
Calcium hydrogen phosphate	0.60	0.60	0.60	0.60
Limestone	0.50	0.50	0.50	0.50
Nutritional composition ^3^				
ME (MJ/kg)	7.34	7.34	7.34	7.34
CP (%DM)	13.61	13.61	13.61	13.61
NDF (%DM)	43.20	43.20	43.20	43.20
ADF (%DM)	22.89	22.89	22.89	22.89
Ca (%DM)	0.47	0.47	0.47	0.47
P (%DM)	0.22	0.22	0.22	0.22
S (%DM)	0.40	0.60	0.80	1.00

^1^ CON = control, LAS = low additional sulfur, MAS = moderate additional sulfur, HAS = high additional sulfur; ^2^ Premix: Fe 12 g/kg, Mn 1 g/kg, Cu 1 g/kg, Zn 11 g/kg, I 30 mg/kg, Se 30 mg/kg, Co 20 mg/kg, Vitamin A 450,000 IU/kg, Vitamin D3 60,000 IU/kg, Vitamin E 2000 mg/kg; ^3^ Metabolic energy was calculated and other components were determined.

**Table 2 animals-11-02545-t002:** Sequences of primer used to analyze gene expression in rumen epithelial tissue by quantitative PCR.

Target	Accession Number	Primer Sequences ^1^	PCR Product Size (bp)
Claudin-1	NM_001001854.2	F: 5′CCGTGCCTTGATGGTGATTGG3′	107
		R: 5′TCTTCTGTGCCTCGTCGTCTTC3′	
Claudin-4	NM_001014391.2	F: 5′ATCGGCAGCAACATCGTCAC3′	107
		R: 5′AGCAGCGAGTCGTACACCTT3′	
Claudin-17	XM_010800919.3	F: 5′GCGTCCGACAAGCCAAGG3′	133
		R: 5′CCAACAAGCAGAGCAATCACAGA 3′	
Occludin	NM_001082433.2	F: 5′GCTACGGCTATGGCTACGGTTA3′	194
		R: 5′CAGGACGGCGGTCACTATTATCA 3′	
TJP1	-	F: 5′ CGGATGGTGCTACAAGTGATGAC3′	194
		R: 5′CGCCTTCTGTGTCTGTGTCTTC3′	
β-actin	NM_173979.3	F: 5′CATCGGCAATGAGCGGTTCC3′	145
		R: 5′CGTGTTGGCGTAGAGGTCCTT3′	

^1^ F and R represent upstream and downstream primers, respectively.

**Table 3 animals-11-02545-t003:** The in vitro rumen fermentation of substrate with different levels of sulfur.

Item ^1^	Treatment									SEM ^2^	*p*-Value	Contrast ^3^	
	0.40%	0.50%	0.60%	0.70%	0.80%	0.90%	1.00%	1.10%	1.20%			P_L_	P_Q_
Gas production dynamic (mL/0.2 g DM)													
GP_24_	55.04 ^ab^	55.72 ^a^	54.22 ^abc^	55.59 ^a^	55.25 ^ab^	52.96^c^	53.29 ^c^	53.97 ^abc^	53.70 ^bc^	0.21	<0.01	<0.01	0.94
GP_48_	62.93 ^abc^	64.18 ^a^	61.88 ^abcd^	63.53 ^ab^	63.42 ^ab^	59.85 ^d^	60.73 ^cd^	62.27 ^abcd^	61.47 ^bcd^	0.31	<0.01	<0.01	0.61
B	63.88 ^abc^	65.36 ^a^	64.97 ^ab^	65.47 ^a^	62.05 ^bcd^	59.48 ^d^	61.90 ^cd^	63.14 ^abc^	60.98 ^cd^	0.48	<0.01	<0.01	0.75
c (h^−1^)	0.094 ^c^	0.096 ^bc^	0.096 ^bc^	0.099 ^abc^	0.101 ^ab^	0.101 ^ab^	0.101 ^ab^	0.104 ^a^	0.106 ^a^	0.001	0.01	<0.01	0.79
24 h fermentation parameter													
NH_3_-N (mg/100 mL)	17.41	25.69	18.44	18.87	22.39	19.02	18.86	15.69	19.81	0.83	0.19	0.29	0.63
TVFA (mmol/L)	47.37 ^a^	53.62 ^a^	46.02 ^ab^	47.77 ^a^	38.8 ^b^	51.89 ^a^	53.57 ^a^	49.99 ^a^	49.66 ^a^	1.12	0.02	0.40	0.22
Acetate (%)	64.01	64.06	63.18	64.51	62.04	63.45	64.10	63.70	62.98	0.24	0.13	0.31	0.59
Propionate (%)	19.06	18.70	19.01	18.52	19.16	19.10	18.54	18.50	19.10	0.17	0.37	0.70	0.67
Isobutyrate (%)	1.36	1.57	1.55	1.61	1.68	1.45	1.69	1.77	1.60	0.11	0.46	0.09	0.45
Butyrate (%)	12.11	12.21	12.67	12.02	13.37	12.53	12.23	12.49	12.76	0.20	0.18	0.24	0.41
Isovalerate (%)	2.67	2.59	2.75	2.50	2.90	2.61	2.49	2.63	2.65	0.09	0.39	0.68	0.82
Valerate (%)	0.79	0.87	0.83	0.85	0.85	0.86	0.95	0.91	0.90	0.02	0.81	0.12	0.82
A/P	3.36	3.43	3.32	3.49	3.24	3.32	3.46	3.45	3.30	0.05	0.19	0.91	0.94
48 h fermentation parameter													
NH_3_-N (mg/100 mL)	26.62	27.99	27.8	29.67	27.16	25.53	28.49	31.56	28.18	0.79	0.86	0.48	0.89
TVFA (mmol/L)	54.67	46.69	50.39	49.15	59.54	49.17	57.50	62.16	62.26	1.82	0.29	0.04	0.27
Acetate (%)	63.11	63.12	62.40	62.53	62.81	63.28	62.55	62.93	62.69	0.18	0.97	0.79	0.75
Propionate (%)	18.50	18.44	18.45	18.43	18.41	18.49	18.31	18.06	18.18	0.06	0.81	0.11	0.51
Isobutyrate (%)	2.11	2.22	2.12	2.12	2.33	2.08	2.40	2.32	2.51	0.07	0.86	0.17	0.57
Butyrate (%)	11.88	11.85	12.32	12.25	12.03	11.98	12.19	11.98	12.13	0.08	0.92	0.67	0.51
Isovalerate (%)	3.48	3.50	3.65	3.63	3.33	3.27	3.43	3.45	3.40	0.05	0.77	0.33	0.90
Valerate (%)	0.92	0.87	1.07	1.03	1.09	0.91	1.12	1.25	1.10	0.04	0.26	0.03	0.83
A/P	3.41	3.42	3.38	3.39	3.41	3.42	3.42	3.48	3.45	0.02	0.93	0.27	0.48

^1^ DM = dry matter; GP_24_ = gas production at 24h; GP_48_ = gas production at 48h; B = asymptotic gas production, c = rate of gas production; NH3-N = ammonia nitrogen; TVFA = total volatile fatty acid; A/P = the ratio between the content of acetate and propionate; ^2^ SEM = standard error of the mean; ^3^ L = linear effect; Q = quadratic effect. Means in the same row, followed by different superscripts (a, b, c and d) are significantly different (*p* < 0.05).

**Table 4 animals-11-02545-t004:** Effect of additional sulfur in the diet on in vitro rumen fermentation parameters.

Item	Treatment				SEM	*p*-Value
	CON	LAS	MAS	HAS		
NH_3_-N (mg/100 mL)	18.63 ^a^	5.98 ^c^	6.88 ^bc^	14.91 ^ab^	1.66	<0.01
TVFA (mmol/L)	41.72 ^b^	38.51 ^b^	39.15 ^b^	52.87 ^a^	1.35	<0.01
Acetate (%)	67.66 ^ab^	67.81 ^a^	66.42 ^b^	66.41 ^b^	0.23	0.03
Propionate (%)	20.18 ^a^	16.09 ^c^	20.03 ^a^	18.14 ^b^	0.38	<0.01
Isobutyrate (%)	0.63 ^b^	1.03 ^a^	0.71 ^b^	0.57 ^b^	0.05	<0.01
Butyrate (%)	9.19 ^c^	13.56 ^a^	11.13 ^b^	13.24 ^a^	0.35	<0.01
Isovalerate (%)	1.70 ^a^	1.30 ^b^	1.33 ^b^	1.18 ^b^	0.06	0.01
Valerate (%)	0.63 ^a^	0.20 ^c^	0.37 ^b^	0.46 ^b^	0.03	<0.01
A/P	3.37 ^bc^	4.24 ^a^	3.33 ^c^	3.69 ^b^	0.08	<0.01

Note: CON-S (%DM) 0.4%; LAS-S (%DM) 0.6%; MAS-S (%DM) 0.8%; HAS-S (%DM) 1.0%. Means in the same row, followed by different superscripts (a, b, and c) are significantly different (*p* < 0.05).

**Table 5 animals-11-02545-t005:** Influence of different sulfur content on the alpha diversity of bacterial communities in the rumen.

Item ^1^	Observed Species	Shannon	Simpson	Ace	Chao	Good’s Coverage
CON	1580.50	6.08	0.0074	1857.33	1882.86	0.9826
LAS	1467.25	5.91	0.0089	1757.12	1770.60	0.9830
MAS	1522.00	5.95	0.0095	1813.51	1836.60	0.9824
HAS	1518.13	6.01	0.0083	1797.46	1821.01	0.9830
SEM	16.36	0.03	0.0006	17.54	18.68	0.0002
*p*-Value	0.10	0.33	0.67	0.25	0.21	0.65

^1^ CON-S (%DM) 0.4%; LAS-S (%DM) 0.6%; MAS-S (%DM) 0.8%; HAS-S (%DM) 1.0%.

**Table 6 animals-11-02545-t006:** Effect of high sulfur diet on the thickness of different cellular stratum of rumen epithelium.

Item ^1^	Treatment ^2^				SEM	*p*-Value
	CON	LAS	MAS	HAS		
Epithelial papilla length/mm	5.24 ^c^	7.33 ^bc^	8.67 ^ab^	10.94 ^a^	0.58	<0.01
spinous layer + basal layer/µm	66.16 ^b^	67.23 ^b^	73.42 ^ab^	80.10 ^a^	1.86	0.02
cuticle + granular layer/µm	16.88	15.01	18.25	18.57	0.62	0.17
Total epithelium/µm	83.04 ^b^	82.24 ^b^	91.67 ^ab^	98.67 ^a^	2.07	0.01

^1^ Total epithelium = cuticle + granular layer + spinous layer + basal layer; ^2^ CON-S (%DM) 0.4%; LAS-S (%DM) 0.6%; MAS-S (%DM) 0.8%; HAS-S (%DM) 1.0%. Means in the same row, followed by different superscripts (a, b, and c) are significantly different (*p* < 0.05).

## Data Availability

All data are presented in the text and tables of this manuscript.

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
