# Peer review of "Effect of High Sulfur Diet on Rumen Fermentation, Microflora, and Epithelial Barrier Function in Steers"

_animals, 2021, doi:10.3390/ani11092545_

Round 1
Reviewer 1 Report
Hydrogen sulfide has been the cause of many accidents in volcanic areas including hot springs, natural gas and oil extraction sites, and labor sites. In recent years, there have also been reports of its use as a suicide tool.
Doesn't hydrogen sulfide have a negative impact on cattle health ?
P2; SRB --- sulfate-reducing bacteria (SRB)
Materials and Methods
You should indicate whether or not animal experiment has been done properly. For example
'All procedures in this study were performed according to the Animal Protection Law based on the Guide for the Care and Use of Laboratory Animals approved by the Ethics Committee of University of Las Palmas de Gran Canaria (No. 2021-0001).'
nine syringes --- Did you use only 9-syringes ? n=1 in each experimental treatment ?
Triplicate gas samples from 1 syringe were not repetition.
P4; (Sigma, USA) (Thermo Scientific, USA) --- Which state are they ?
(Life Technologies, Inc.) --- Where is company ?
P5; (Thermo Fisher Scientific, Madison, Wisconsin, USA) --- (Thermo Fisher Scientific, Madison, WI, USA)
California, USA) --- CA, USA)
according to the 2–△△CT method --- OK ?
Figure 2. How do you include 0% ? Did you include O% ?
Concentration(%) varies greatly depending on the total amount of gas produced. Isn't it possible to make an accurate comparison only with the amount of methane produced (ml)?
P7; doses on NH3-N and VFA --- subscript
Total Volatile Fatty Acids --- total volatile fatty acids
Where are the in vivo results ?
individual volatile fatty acid (VFA) concentration --- individual VFA concentration
P19. Where is 5. Conclusions ?
Table 3. (a and b) --- (a, b, c and d)
Figure 6. What do the horizontal bars and **** mean, is it only the difference between CON and HAS ?
Table 6. What are a, b and c ?
Table 5, 6, Figure 3, 6, 7, 11. What are CON, LAS, MAS and HAS ?
Figure 8, 10. What are S1, S2 and S3 ?
Figure 11. What are TJP-1, OCLN, CLDN-1, CLDN-4 and CLDN-17 ?
Author Response
Please see our responses in the attachment. The changes we made to the text of the manuscript were shown in blue for the ease of lacating them.

Reviewer 2 Report
Comments to the Authors of manuscript number: animals-1344695 entitled “Effect of High Sulfur Diet on Rumen Fermentation, Microflora and Epithelial Barrier Function in Steers”.
Authors have examined many issues related to the rumen physiology in the case of the increased level of sulfur in the diet of polygastric animals. The methodology used is very good, but there is many gaps which should be explained.
It is very useful study. It can be published after substantial correction.
- Simple summary: abbreviation should be explained
Abstract:
- to what does it relate: epithelial barrier- it should be clarified
- It is not clear: if sulphur did not affect the rumen fermentation how butyric acid concentration increases?
- SRB?
Introduction:
- DDGS-?
- SRB- ?
- H2S or sulfur? The title indicates that sulfur.
- It is not mechanism but the result of the high-sulfur diet
The part 2.2
- Where are presented the components of artificial saliva?
The part of 2.2
- How the levels of sulfur were chosen?
- How the level of sulfur was determined in basal diet?
- If sodium sulfate was given as an examined diet, how the level of Na was rebalanced in the control diet? It should be explained
- Was the sulfur content in plants of basal diet in the examined groups determined? how much sulfur was in the plants in the tested diet? It should be given.
- Which level of sulfur relates to what treatment should be clearly indicated.
Table 1
- What was sodium level in the control group?
- How all these components were determined?
The part of 2.3.2
- How the samples of rumen epithelium were collected? Were animals under anesthesia?
It should be explained.
The part of 2.5
- Was the number of papillae counted? Its number and shape influence the surface of the epithelium. Was the surface of rumen epithelium determined?
Table 2
- Why only proteins of TJ were chosen? It should be explained.
The part of 3.2
- It is known that bacteria, protozoa and fungi are among the microbes of the rumen, and they influence to each other. Why the number of protozoa were not calculated? Protozoa are the important source of protein for host. It should be explained.
Figure 10
- There is a lack of description of this figure. It is not know what kind of cell`structure is visible. It should be explained.
Discussion
- Methane is also produced by a specific kind of bacteria present in rumen. They are archaea.
CO2 ± 4H2 → CH4 ± 2H2O
What about they?
The concentration of hydrogen that creates a pressure not exceeding 1 kP promotes the development of cellulolytic bacteria and the maintenance of acetic acid fermentation
- The part about IBD should be omitted.
- The fermentation influences the shape and number of papillae. The size depends on the fermentability of the feed. It should be determined and discussed.
- Why methionine, cysteine, homocysteine, and taurine concentration (the 4 common sulfur-containing amino acids) in the blood were not determined?
Author Response
Please see our responses in the attachment. The changes we made to the text of the manuscript were shown in green for the ease of lacating them.

Round 2
Reviewer 1 Report
I have no further comments.
Author Response
Thank you for your kindly reply.
Reviewer 2 Report
Comments to the Authors of manuscript number: animals-1344695 entitled “Effect of High Sulfur Diet on Rumen Fermentation, Microflora and Epithelial Barrier Function in Steers”.
Authors have tried to answer, but I did not expect answer only in cover letter to reviewer but I expected that Authors have explained all these issues in the text.
Moreover, the introduction colourful arrows in e.g. Figure 10 does not indicate that the description of this figure was improved. There still is lack proper correction. What do these arrows indicate. Everything should be explained and described. All answer should be introduced into the text in the proper places.
Round 3
Reviewer 2 Report
Dear Authors,
please correct properly?
1.Figures are still uncler. What it means?:
"A, CON group-S (%DM) 0.4%; B, LAS group-S (%DM) 0.6%; C, MAS group-S 383 (%DM) 0.8%; D, HAS group-S (%DM) 1.0%."? What is %DM?
each figure should be described in the manner which allows to understand what it presents.
2. Figure 7 - layers of papille should be described. No one can know histology. The description should be placed under the figure.
3. Figure 8 and 9. Except for the information that it is "Scanning electron microscopy of papillae surface of different groups" somone does not know can see. The description should be placed under each figure
4. Figure 10. yellow blue or other arrows ...what is pointed by they at? This description should be under the figure.
Author Response
Dear reviewer,
Sorry for the misunderstanding, we described the meaning of CON, LAS, MAS and HAS in the Materials and Methods part, which meant the sulfur contents in the diet were 0.4%, 0.6%,0.8% and 1.0%, respectively. Moreover, “% DM” meant that the compositions were presented as dried matter basis. We made changes to the legends to make the figure more understandable.